# DEEP ReLU NETWORKS PRESERVE
# EXPECTED LENGTH

**Boris Hanin** *
Dept. of Operations Research
& Financial Engineering
Princeton University
Princeton, NJ 08544 USA
`bhanin@princeton.edu`

**Ryan Jeong** *
Dept. of Mathematics
University of Pennsylvania
Philadelphia, PA 19104 USA
`rsjeong@sas.upenn.edu`

**David Rolnick** *
School of Computer Science
McGill University
Montréal, QC H3A 0G4 Canada
`drolnick@cs.mcgill.ca`

## ABSTRACT

Assessing the complexity of functions computed by a neural network helps us understand how the network will learn and generalize. One natural measure of complexity is how the network distorts length – if the network takes a unit-length curve as input, what is the length of the resulting curve of outputs? It has been widely believed that this length grows exponentially in network depth. We prove that in fact this is not the case: the expected length distortion does not grow with depth, and indeed shrinks slightly, for ReLU networks with standard random initialization. We also generalize this result by proving upper bounds both for higher moments of the length distortion and for the distortion of higher-dimensional volumes. These theoretical results are corroborated by our experiments.

## 1 INTRODUCTION

The utility of deep neural networks ultimately arises from their ability to learn functions that are sufficiently complex to fit training data and yet simple enough to generalize well. Understanding the precise sense in which functions computed by a given network have low or high complexity is therefore important for studying when the network will perform well. Despite the fundamental importance of this question, our mathematical understanding of the functions expressed and learned by different neural network architectures remains limited.

A popular way to measure the complexity of a neural network function is to compute how it distorts lengths. This may be done by considering a set of inputs lying along a curve and measuring the length of the corresponding curve of outputs. It has been claimed in prior literature that in a ReLU network this *length distortion* grows exponentially with the network's depth Price & Tanner (2019); Raghu et al. (2017), and this has been used as a justification of the power of deeper networks. We prove that, in fact, for networks with the typical initialization used in practice, expected length distortion does not grow at all with depth.

Our main contributions are:

1. We prove that for ReLU networks initialized with the usual 2/fan-in weight variance, the expected length distortion does not grow with depth at initialization, actually decreasing slightly with depth (Thm. 3.1) and exhibiting an interesting width dependency.

2. We prove bounds on higher moments of the length distortion, giving upper bounds that hold with high probability (Thm. 4.1). We also obtain similar results for the distortion in the volume of higher-dimensional manifolds of inputs (Thm. 4.2).

3. We empirically verify that our theoretical results accurately predict observed behavior for networks at initialization, while previous bounds are loose and fail to capture subtle architecture dependencies.

It is worth explaining why our conclusions differ from those of Price & Tanner (2019); Raghu et al. (2017). First, prior authors prove only *lower bounds* on the expected length distortion, while we use

---
*Equal contribution

different methodology to calculate tight upper bounds, allowing us to say that the expected length distortion *does not grow* with depth. As we show in Thm. 5.1 and Fig. 1, the prior bounds are in fact quite loose. Second, Thm. 3(a) in Raghu et al. (2017) has a critical typo[1], which has unfortunately been perpetuated in Thm. 1 of Price & Tanner (2019). Namely, the "2" was omitted in the following statement (paraphrased from the original):

$$\mathbb{E}\left[\text{length distortion}\right] = \Omega\left[\left(\frac{\sigma_w\sqrt{\text{width}}}{2\sqrt{\text{width}+1}}\right)^{\text{depth}}\right],$$

where $\sigma_w^2/\text{width}$ is the variance of the weight distribution. As we prove in Thm. 3.1, leaving out the 2 makes the statement false, incorrectly suggesting that length distortion explodes with depth for the standard He initialization $\sigma_w = \sqrt{2}$.

Finally, prior authors drew the conclusion that length distortion grows exponentially with depth by considering the behavior of ReLU networks with unrealistically large weights. If one multiplies by $C$ the weights and biases of a ReLU network, one multiplies the length distortion by $C^{\text{depth}}$, so it should come as no surprise that there exist settings of the weights for which the distortion grows (or decays) exponentially with depth. The value of our results comes in analyzing the behavior specifically at He initialization ($\sigma_w = \sqrt{2}$) He et al. (2015). This initialization is the one used in practice, since this is the weight variance that must be used if the outputs Hanin & Rolnick (2018) and gradients Hanin (2018) are to remain well-controlled at init. In the present work, we show that this is also the correct initialization for the expected length distortion to remain well-behaved.

## 2 RELATED WORK

A range of complexity measures for functions computed by deep neural networks have been considered in the literature, dividing the prior work into at least three categories. In the first, the emphasis is on *worst-case* (or *best-case*) scenarios – what is the maximal possible complexity of functions computed by a given network architecture. These works essentially study the expressive power of neural networks and often focus on showing that deep networks are able to express functions that cannot be expressed by shallower networks. For example, it has been shown that it is possible to set the weights of a deep ReLU network such that the number of linear regions computed by the network grows exponentially in the depth Daniely (2017); Eldan & Shamir (2016); Montúfar et al. (2014); Telgarsky (2015; 2016). Other works consider the degree of polynomials approximable by networks of different depths Lin et al. (2017); Rolnick & Tegmark (2018) and the topological invariants of networks Bianchini & Scarselli (2014).

While such work has sometimes been used to explain the utility of different neural network architectures (especially deeper ones), a second strand of prior work has shown that a significant gap can exist between the functions expressible by a given architecture and those which may be learned in practice. Such *average-case* analyses have indicated that some readily expressible functions are provably difficult to learn Shalev-Shwartz et al. (2017) or vanishingly unlikely for random networks Hanin & Nica (2020); Hanin & Rolnick (2019; 2020), and that some functions learned more easily by deep architectures are nonetheless expressible by shallow ones Ba & Caruana (2014). (While here we consider neural nets with ReLU activation, it is worth noting that for arithmetic circuits, worst-case and average-case scenarios may be more similar – in both scenarios, a significant gap exists between the matricization rank of the functions computed by deep and shallow architectures Cohen et al. (2016).) As noted earlier, average-case analyses in Price & Tanner (2019); Raghu et al. (2017) provided lower bounds on expected length distortion, while Poole et al. (2016) presented a similar analysis for the curvature of output trajectories.

Finally, a number of authors have sought complexity measures that either empirically or theoretically correlate with generalization Jiang et al. (2019). Such measures have been based on classification margins Bartlett et al. (2017), network compressibility Arora et al. (2018), and PAC-Bayes considerations Dziugaite & Roy (2017).

---

[1]We have confirmed this in personal correspondence with the authors, and it is simple to verify – the typo arises in the last step of the proof given in the Supplementary Material of Raghu et al. (2017).

## 3 EXPECTED LENGTH DISTORTION

### 3.1 MOTIVATION

While neural networks are typically overparameterized and trained with little or no explicit regularization, the functions they learn in practice are often able to generalize to unseen data. This phenomenon indicates an implicit regularization that causes these learned functions to be surprisingly simple.

Why this occurs is not well-understood theoretically, but there is a high level intuition. In a nutshell, randomly initialized neural networks will compute functions of low complexity. Moreover, as optimization by first order methods is a kind of greedy local search, network training is attracted to minima of the loss that are not too far from the initialization and hence will still be well-behaved.

While this intuition is compelling, a key challenge in making it rigorous is to devise appropriate notions of complexity that are small throughout training. The present work is intended to provide a piece of the puzzle, making precise the idea that, at the start of training, neural networks compute tame functions. Specifically, we demonstrate that neural networks at initialization have low distortion of length and volume, as defined in §3.2.

An important aspect of our analysis is that we study networks in typical, average-case scenarios. A randomly initialized neural network could, in principle, compute any function expressible by a network of that architecture, since the weights might with low probability take on any set of values. Some settings of weights will lead to functions of high complexity, but these settings may be unlikely to occur in practice, depending on the distribution over weights that is under consideration. As prior work has emphasized Price & Tanner (2019); Raghu et al. (2017), atypical distributions of weights can lead to exponentially high length distortion. We show that, in contrast, for deep ReLU networks with the standard initialization, the functions computed have low distortion in expectation and with high probability.

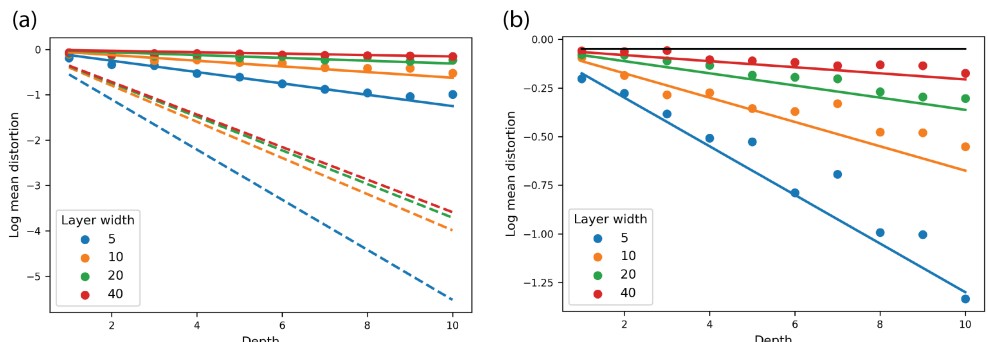

Figure 1: Mean length distortion as a function of depth, for randomly initialized ReLU networks of varying architecture. As described in Thm. 3.1, distortion not only fails to grow, but shrinks with depth, especially for networks of small width. (a) compares the predictions of our Thm. 5.1 (solid lines) to the lower bounds proven in prior work Raghu et al. (2017) (dashed lines) and the true empirical means (colored dots), which closely track our predictions. (b) zooms in on the upper part of (a), showing the empirical mean length distortion as a function of depth for different widths. The horizontal black line is $y = \frac{\Gamma(3)}{\Gamma(2.5)\sqrt{2.5}}$, the mean length distortion we predict when $n_0 = n_L$ in the limit of infinite width for any fixed depth. In each network, width is constant in all hidden layers, while input and output dimension are both 5. The input curve is a fixed line segment of unit length. Length distortion is calculated for 500 different initializations of the weights and biases of the network (the weight variance is 2/fan-in). For further experimental details, see Appendix B.

### 3.2 DEFINITIONS

Let $L \geq 1$ be a positive integer and fix positive integers $n_0, \ldots, n_L$. We consider a fully connected feed-forward ReLU network $\mathcal{N}$ with input dimension $n_0$, output dimension $n_L$, and hidden layer

widths $n_1, \dots, n_{L-1}$. Suppose that the weights and biases of $\mathcal{N}$ are *independent* and Gaussian with the weights $W_{ij}^{(\ell)}$ between layers $\ell - 1$ and $\ell$ and biases $b_j^{(\ell)}$ in layer $\ell$ satisfying:

$$W_{ij}^{(\ell)} \sim G\left(0, 2/n_{\ell-1}\right), \qquad b_j^{(\ell)} \sim G(0, C_b). \tag{1}$$

Here, $C_b > 0$ is any fixed constant and $G(\mu, \sigma^2)$ denotes a Gaussian with mean 0 and variance $\sigma^2$. For any $M \subseteq \mathbb{R}^{n_0}$, we denote by $\mathcal{N}(M) \subseteq \mathbb{R}^{n_L}$ the (random) image of $M$ under the map $x \mapsto \mathcal{N}(x)$. Our primary object of the study will be the size of the output $\mathcal{N}(M)$ relative to that of $M$. Specifically, when $M$ is a 1-dimensional curve, we define

$$\textit{length distortion} = \frac{\text{len}(\mathcal{N}(M))}{\text{len}(M)}.$$

Note that while *a priori* this random variable depends on $M$, we will find that its statistics depend only on the architecture of $\mathcal{N}$ (see Thms. 3.1 and 5.1).

## 3.3 Results

We prove that the expected length distortion does not grow with depth – in fact, it slightly decreases. Our result may be informally stated thus (for a formal statement and proof, see Thm. 5.1 and App. C.):

**Theorem 3.1** (Length distortion: Mean; Informal Statement of Theorem 5.1). *Consider a ReLU network of depth $L$, input dimension $n_0$, output dimension $n_L$, and hidden layer widths $n_\ell$, with weights given by standard He normal initialization He et al. (2015). The expected length distortion is upper bounded by $\sqrt{n_L/n_0}$. More precisely:*

$$\mathbb{E}\left[\textit{length distortion}\right] \approx C \left(\frac{n_L}{n_0}\right)^{1/2} \exp\left[-\frac{5}{8} \sum_{\ell=1}^{L-1} \frac{1}{n_\ell}\right], \qquad C := \frac{\Gamma\left(\frac{n_L+1}{2}\right)}{\Gamma\left(\frac{n_L}{2}\right)\sqrt{\frac{n_L}{2}}} \approx 1.$$

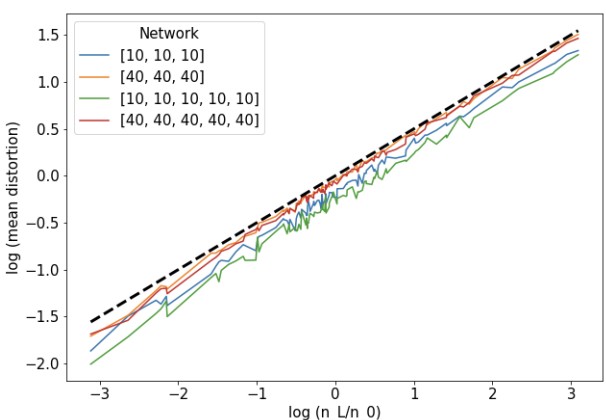

Figure 2: Mean length distortion as a function of the ratio of output to input dimension, for ReLU networks with various architectures (e.g. $[10, 10, 10]$ denotes three hidden layers, each of width 10). All networks are randomly initialized as in Figure 1. We sample 100 pairs of input dimension $n_0$ and output dimension $n_L$, each at most 50, such that the ratio of output to input dimension is distinct for each such pair. For each pair, 200 different network initializations are tested and the resulting length distortion is calculated; the log of the empirical mean is plotted. The dashed black line plots $\log(y) = \frac{1}{2}\log(x)$, the approximate prediction by Theorem 3.1.

Our experiments align with the theorem. In Figure 1, we compute the empirical mean of the length distortion for randomly initialized deep ReLU networks of various architectures with fixed input and output dimension. The figure confirms three of our key predictions: (1) the expected length distortion does not grow with depth, and actually decreases slightly; (2) the decrease happens faster for narrower networks (since $1/n_\ell$ is larger for smaller $n_\ell$); and (3) for equal input and output dimension $n_0 = n_L$, there is an upper bound of 1 (in fact, the tighter upper bound of $C$ is approximately 0.9515 for $n_L = 5$, as shown in the figure. The figure also shows the prior bounds in Raghu et al. (2017) to be quite loose. For further experimental details, see Appendix B.

In Fig. 2, we instead fix the set of hidden layer widths, but vary input and output dimensions. The results confirm that indeed the expected length distortion grows as the square root of the ratio of output to input dimension.

Note that our theorem also applies for initializations other than He normal; the result in such cases is simply less interesting. Suppose we initialize the weights in each layer $\ell$ as i.i.d. normal with

variance $2c^2/n_{\ell-1}$ instead of $2/n_{\ell-1}$. This is equivalent to taking a He normal initialization and multiplying the weights by $c$. Multiplying the weights and biases in a depth-$L$ ReLU network by $c$ simply multiplies the output (and hence the length distortion) by $c^L$, so the expected distortion is $c^L$ times the value given in Thm. 3.1. This emphasizes why an exponentially large distortion necessarily occurs if one sets the weights too large.

### 3.4 INTUITIVE EXPLANATION

The purpose of this section is to give an intuitive but technical explanation for why, in Theorem 3.1, we find that the distortion $\mathrm{len}(\mathcal{N}(M))/\mathrm{len}(M)$ of the length of a $1D$ curve $M$ under the neural network $\mathcal{N}$ is typically not much larger than 1, even for deep networks.

Our starting point is that, near a given point $x \in M$, the length of the image under $\mathcal{N}$ of a small portion of $M$ with length $dx$ is approximately given by $||J_x u|| \, dx$, where $J_x$ is the input-output Jacobian of $\mathcal{N}$ at the input $x$ and $u$ is the unit tangent vector to $M$ at $x$. Prior work (see Fact 7.2 in Allen-Zhu et al. (2019)).

In fact, in Lemma C.1 of the Supp. Material, we use a simple argument to give upper bounds on the moments of $\mathrm{len}(\mathcal{N}(M))/\mathrm{len}(M)$ in terms of the moments of the norm $||J_x u||$ of the Jacobian-vector product $J_x u$.

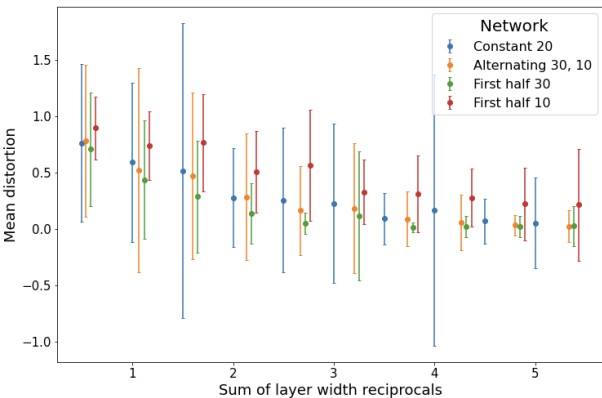

Figure 3: Length distortion as a function of $\sum_{\ell=1}^{L} n_\ell^{-1}$, showing both mean and standard deviation across initializations. We test several types of network architecture – with constant width 20, alternating between widths 30 and 10, and with the first (respectively, second) half of the layers of width 30 and the rest width 10. Each architecture type is tested for several depths. For each such network, we use $n_0 = n_L = 5$ and compute length distortion for 100 initializations on a fixed line segment. As predicted in Thm. 4.1, the mean length distortion decreases with the sum of width reciprocals. Empirical standard deviation does not, in general, increase with greater depth, remaining modest throughout, as is consistent with the upper bound on variance in Thm. 4.1.

Thus, upper bounds on the length distortion reduce to studying the Jacobian $J_x$, which in a network with $L$ hidden layers can be written as a product $J_x = J_{L,x} \cdots J_{1,x}$ of the layer $\ell - 1$ to layer $\ell$ Jacobians $J_{\ell,x}$. In a worst-case analysis, the left singular vectors of $J_{\ell,x}$ would align with the right singular vectors of $J_{\ell+1,x}$, causing the resulting product $J_x$ to have a largest singular value that grows exponentially with $L$. However, at initialization, this is provably not the case with high probability. Indeed, the Jacobians $J_{\ell,x}$ are independent (see Lemma C.3) and their singular vectors are therefore incoherent. We find in particular (Lemma C.2) the following equality in distribution:

$$||J_x u|| \stackrel{d}{=} \prod_{\ell=1}^{L} ||J_{\ell,x} e_1||,$$

where $e_1$ is the first standard unit vector and the terms in the product are independent. On average each term in this product is close to 1:

$$\mathbb{E}\left[||J_{\ell,x} e_1||\right] = 1 + O(n_\ell^{-1}).$$

This is a consequence of initializing weights to have variance 2/fan-in. Put together, the two preceding displayed equations suggest writing

$$||J_x u|| = \exp\left[\sum_{\ell=1}^{L} \log(||J_{\ell,x} e_1||)\right]$$

The terms being summed in the exponent are independent and the argument of each logarithm scales like $1 + O(n_\ell^{-1})$. With probability $1 - \delta$ we have for all $\ell$ that $\log(||J_{\ell,x} e_1||) \le c n_\ell^{-1}$, where $c$ is a

constant depending only on $\delta$. Thus, all together,

$$||J_x u|| \leq \exp \left[ c \sum_{\ell=1}^{L} n_\ell^{-1} \right]$$

with high probability. In particular, in the simple case where $n_\ell$ are proportional to a single large constant $n$, we find the typical size of $||J_x u||$ is exponential in $L/n$ rather than exponential in $L$, as in the worst case. If $\mathcal{N}$ is wider than it is deep, so that $L/n$ is bounded above, the typical size of $||J_x u||$ and hence of $\text{len}(\mathcal{N}(M))/\text{len}(M)$ remains bounded. Theorem 5.1 makes this argument precise. Moreover, let us note that articles like Daniely (2017); Giryes et al. (2016); Poole et al. (2016) show low distortion of angles and volumes in very wide networks. Our results, however, apply directly to more realistic network widths.

## 4 Further Results

### 4.1 Higher moments

We have seen that the mean length distortion is upper-bounded by 1, and indeed by a function of the architecture that decreases with the depth of the network. However, a small mean is insufficient by itself to ensure that typical networks have low length distortion. For this, we now show that in fact all moments of the length distortion are well-controlled. Specifically, the variance is bounded above by the ratio of output to input dimension, and higher moments are upper-bounded by a function that grows very slowly with depth. Our results may be informally stated thus:

**Theorem 4.1** (Length distortion: Higher moments). *Consider, as before, a ReLU network of depth $L$, input dimension $n_0$, output dimension $n_L$, and hidden layer widths $n_\ell$, with weights given by He normal initialization. We have the following bounds on higher moments of the length distortion:*

$$\text{Var}[\text{length distortion}] \leq \frac{n_L}{n_0} \quad \text{and} \quad \mathbb{E}\left[(\text{length distortion})^m\right] \leq \left(\frac{n_L}{n_0}\right)^{\frac{m}{2}} \exp\left[ cm^2 \sum_{\ell=1}^{L} n_\ell^{-1} \right]$$

*for some universal constant $c > 0$.*

A formal statement and proof of this result are given in Theorem 5.1 and Appendix C.

We consider the mean and variance of length distortion for a wide range of network architectures in Figure 3. The figure confirms that variance is modest for all architectures and does not increase with depth, and that mean length distortion is consistently decreasing with the sum of layer width reciprocals, as predicted by Theorem 4.1.

### 4.2 Higher-dimensional volumes

Another natural generalization to consider is how ReLU networks distort higher-dimensional volumes: Given a $d$-dimensional input $M$, the output $\mathcal{N}(M)$ will in general also be $d$-dimensional, and we can therefore consider the *volume distortion* $\text{vol}_d(\mathcal{N}(M))/\text{vol}_d(M)$. The theorems presented in §3.3 when $d = 1$ can be extended to all $d$, and the results may be informally stated thus:

**Theorem 4.2** (Volume distortion). *Consider a ReLU network of depth L, input dimension $n_0$, output dimension $n_L$, and hidden layer widths $n_\ell$, with weights given by He normal initialization. Both the squared mean and the variance of volume distortion are upper-bounded by:*

$$\left(\frac{n_L}{n_0}\right)^d \exp\left[ -\binom{d}{2} \sum_{\ell=1}^{L} n_\ell^{-1} \right].$$

A formal statement and proof of this result are given in Theorem 5.2 and Appendix D.

## 5 Formal Statements

In this section, we provide formal statements of our results. Full proofs are given in the appendices.

## 5.1 ONE-DIMENSIONAL MANIFOLDS

Fix a smooth one-dimensional manifold $M \subseteq \mathbb{R}^{n_0}$ (i.e. a curve) with unit length. Each point on $M$ represents a possible input to our ReLU network $\mathcal{N}$, which we recall has input dimension $n_0$, output dimension $n_L$, and hidden layer widths $n_1, \ldots, n_{L-1}$. The function $x \mapsto \mathcal{N}(x)$ computed by $\mathcal{N}$ is continuous and piecewise linear. Thus, the image $\mathcal{N}(M) \subseteq \mathbb{R}^{n_L}$ of $M$ under this map is a piecewise smooth curve in $\mathbb{R}^{n_L}$ with a finite number of pieces, and its length $\mathrm{len}(\mathcal{N}(M))$ is a random variable. Our first result states that, provided $\mathcal{N}$ is wider than it is deep, this distortion is small with high probability at initialization.

**Theorem 5.1.** *Let $\mathcal{N}$ be a fully connected ReLU network with input dimension $n_0$, output dimension $n_L$, hidden layer widths $n_1, \ldots, n_{L-1}$, and independent centered Gaussian weights/biases with the weights having variance $2/\text{fan-in}$ (as in (1)). Let $M$ be a $1$-dimensional curve of unit length in $\mathbb{R}^{n_0}$. Then, the mean length $\mathbb{E}\left[\mathrm{len}(\mathcal{N}(M))\right]$ equals*

$$\left(\frac{n_L}{n_0}\right)^{1/2} \frac{\Gamma\left(\frac{n_L+1}{2}\right)}{\Gamma\left(\frac{n_L}{2}\right)\left(\frac{n_L}{2}\right)^{1/2}} \times \exp\left[-\frac{5}{8}\sum_{\ell=1}^{L-1} n_\ell^{-1} + O\left(\sum_{\ell=1}^{L-1} n_\ell^{-2}\right)\right], \tag{2}$$

*where implied constants in $O(\cdot)$ are universal and $\Gamma(\cdot)$ denotes the Gamma function. Moreover:*

$$\mathrm{Var}[\mathrm{len}(\mathcal{N}(M))] \leq \frac{n_L}{n_0}. \tag{3}$$

*Finally, there exist universal constants $c_1, c_2 > 0$ such that if $m < c_1 \min\{n_1, \ldots, n_{L-1}\}$, then*

$$\mathbb{E}\left[(\mathrm{len}(\mathcal{N}(M)))^m\right] \leq \left(\frac{n_L}{n_0}\right)^{m/2} \exp\left[c_2 m^2 \sum_{\ell=1}^{L} n_\ell^{-1}\right]. \tag{4}$$

Theorem 5.1 is proved in §C. Several comments are in order. First, we have assumed for simplicity that $M$ has unit length. For curves of arbitrary length, both the equality (2) and the bounds (3) and (4) all hold and are derived in the same way provided $\mathrm{len}(\mathcal{N}(M))$ is replaced by the distortion $\mathrm{len}(\mathcal{N}(M))/\mathrm{len}(M)$ per unit length.

Second, in the expression (2), note that the exponent tends to 0 as $n_\ell \to \infty$ for any fixed $L$. More precisely, when $n_\ell = n$ is large and constant, it scales as $-5L/8n$. This shows that, for fixed $n_0, n_L, n$, the mean $\mathbb{E}\left[\mathrm{len}(\mathcal{N}(M))\right]$ is actually decreasing with $L$. This somewhat surprising phenomenon is borne out in Fig. 1 and is a consequence of the fact that wide fully connected ReLU nets are strongly contracting (see e.g. Thms. 3 and 4 in Giryes et al. (2016)).

Third, the pre-factor $(n_L/n_0)^{1/2}$ encodes the fact that, on average over initialization, for any vector of inputs $x \in \mathbb{R}^{n_0}$, we have (see e.g. Cor. 1 in Hanin & Rolnick (2018))

$$\mathbb{E}\left[\|\mathcal{N}(x)\|^2\right] \approx \frac{n_L}{n_0}\|x\|^2, \tag{5}$$

where the approximate equality becomes exact if the bias variance $C_b$ is set to 0 (see (1)). In other words, at the start of training, the network $\mathcal{N}$ rescales inputs by an average factor $(n_L/n_0)^{1/2}$. This overall scaling factor also dilates $\mathrm{len}(\mathcal{N}(M))$ relative to $\mathrm{len}(M)$.

Fourth, we stated Theorem 5.1 for Gaussian weights and biases (see (1)), but the result holds for any weight distribution that is symmetric around 0, has variance $2/\text{fan-in}$, and has finite higher moments. The only difference is that the constant $5/8$ may need to be slightly enlarged (e.g. around (19)).

Finally, all the estimates (2), (3), and (4) are consistent with the statement that, up to the scaling factor $(n_L/n_0)^{1/2}$, the random variable $\mathrm{len}(\mathcal{N}(M))$ is bounded above by a log-normal distribution:

$$\mathrm{len}(\mathcal{N}(M)) \leq \left(\frac{n_L}{n_0}\right)^{1/2} \exp\left[\frac{1}{2}G\left(\frac{\beta}{2}, \beta\right)\right], \text{ where } \beta = c\sum_{\ell=1}^{L} n_\ell^{-1} \text{ and } c \text{ is a fixed constant.}$$

## 5.2 HIGHER-DIMENSIONAL MANIFOLDS

The results in the previous section generalize to the case when $M$ has higher dimension. To consider this case, fix a positive integer $d \leq \min\{n_0, \ldots, n_L\}$ and a smooth $d-$dimensional manifold $M \subseteq$

$\mathbb{R}^{n_0}$ of unit volume $\mathrm{vol}_d(M) = 1$. Note that if $d > \min\{n_0, \ldots, n_L\}$, then $\mathcal{N}(M)$ is at most $(d-1)$-dimensional and its $d$-dimensional volume vanishes. Its image $\mathcal{N}(M) \subseteq \mathbb{R}^{n_L}$ is a piecewise smooth manifold of dimension at most $d$. The following result, proved in §D, gives an upper bound on the typical value of the $d-$dimensional volume of $\mathcal{N}(M)$.

**Theorem 5.2.** *Let $\mathcal{N}$ be a fully connected ReLU network with input dimension $n_0$, output dimension $n_L$, hidden layer widths $n_1, \ldots, n_{L-1}$ and independent centered Gaussian weights/biases with the variance of the weights given by $2/\text{fan-in (see (1))}$. Let $M$ be a $d$-dimensional smooth submanifold of $\mathbb{R}^{n_0}$ with unit volume and $d \le \min\{n_0, \ldots, n_L\}$. Then, both the squared mean and the variance of the $d-$dimensional volume $\mathrm{vol}_d(\mathcal{N}(M))$ of $\mathcal{N}(M)$ is bounded above by*

$$\left(\frac{n_L}{n_0}\right)^d \exp\left[-\binom{d}{2}\sum_{\ell=1}^{L} n_\ell^{-1}\right] \tag{6}$$

## 6  PROOF SKETCH

The purpose of this section is to explain the main steps to obtaining the mean and variance estimates (2) and (3) from Theorem 5.1. In several places, we will gloss over some mathematical subtleties that we deal with in the detailed proof of Theorem 5.1 given in Appendix C. We will also content ourselves here with proving a slightly weaker estimate than (2) and show instead simply that

$$\mathbb{E}\left[\mathrm{len}(\mathcal{N}(M))\right] \le \left(\frac{n_L}{n_0}\right)^{1/2} \qquad \text{and} \qquad \mathrm{Var}\left[\mathrm{len}(\mathcal{N}(M))\right] \le \frac{n_L}{n_0}. \tag{7}$$

We refer the reader to Appendix C for a derivation of the more refined result stated in Theorem 5.1. Since we have $\mathbb{E}[X]^2 \le \mathbb{E}[X^2]$ and $\mathrm{Var}[X] \le \mathbb{E}[X^2]$, both estimates in (7) follow from

$$\mathbb{E}\left[\mathrm{len}(\mathcal{N}(M))^2\right] \le \frac{n_L}{n_0}. \tag{8}$$

To obtain this bound, we begin by relating moments of $\mathrm{len}(\mathcal{N}(M))$ to those of the input-output Jacobian of $\mathcal{N}$ at a single point for which we need some notation. Namely, let us choose a unit speed parameterization of $M$; that is, fix a smooth mapping

$$\gamma : \mathbb{R} \to \mathbb{R}^{n_0}, \qquad \gamma(t) = (\gamma_1(t), \ldots, \gamma_{n_0}(t))$$

for which $M$ is the image under $\gamma$ of the interval $[0,1]$. That $\gamma$ has unit speed means that for every $t \in [0,1]$ we have $||\gamma'(t)|| = 1$, where $\gamma'(t) := (\gamma_1'(t), \ldots, \gamma_{n_0}'(t))$. Then, the mapping $\Gamma := \mathcal{N} \circ \gamma$ (for $\Gamma : \mathbb{R} \to \mathbb{R}^{n_L}$) gives a parameterization of the curve $\mathcal{N}(M)$. Note that this parameterization is not unit speed. Rather the length of $\mathcal{N}(M)$ is given by

$$\mathrm{len}(\mathcal{N}(M)) = \int_0^1 ||\Gamma'(t)|| \, dt. \tag{9}$$

Intuitively, the integrand $||\Gamma'(t)|| \, dt$ computes, at a given $t$, the length of $\Gamma([t, t+dt])$ as $dt \to 0$. The following Lemma uses (9) to bound the moments of $\mathrm{len}(\mathcal{N}(M))$ in terms of the moments of the norm of the input-output Jacobian $J_x$, defined for any $x \in \mathbb{R}^{n_0}$ by

$$J_x := \left(\frac{\partial \mathcal{N}_i}{\partial x_j}(x)\right)_{\substack{1 \le i \le n_L \\ 1 \le j \le n_0}}, \tag{10}$$

where $\mathcal{N}_i$ is the $i$-th component of the network output.

**Lemma 6.1.** *We have*

$$\mathbb{E}\left[\mathrm{len}(\mathcal{N}(M))^2\right] \le \int_0^1 \mathbb{E}\left[\left|\left|J_{\gamma(t)}\gamma'(t)\right|\right|^2\right] dt. \tag{11}$$

*Sketch of Proof.* Taking powers in (9) and interchanging the expectation and integrals, we obtain

$$\mathbb{E}\left[\mathrm{len}(\mathcal{N}(M))^2\right] = \int_0^1 \int_0^1 \mathbb{E}\left[||\Gamma'(t_1)|| \, ||\Gamma'(t_2)||\right] dt_1 dt_2. \tag{12}$$

Applying the inequality $ab \leq \frac{1}{2}(a^2 + b^2)$, for $a, b \in \mathbb{R}$, to the integrand in (12), we conclude

$$\mathbb{E}\left[\text{len}(\mathcal{N}(M))^2\right] \leq \int_0^1 \mathbb{E}\left[||\Gamma'(t)||^2\right] dt. \tag{13}$$

The chain rule yields $\Gamma'(t) = J_{\gamma(t)}\gamma'(t)$. Substituting this into (13) completes the proof. $\square$

Lemma 6.1, while elementary, appears to be new, allowing us to bound global quantities such as length in terms of local quantities such as the moments of $||J_x u||$. In particular, having obtained the expression (11), we have reduced bounding the second moment of $\text{len}(\mathcal{N}(M))$ to bounding the second moment of the random variable $||J_x u||$ for a fixed $x \in \mathbb{R}^{n_0}$ and unit vector $u \in \mathbb{R}^{n_0}$. This is a simple matter since in our setting the distribution of weights and biases is Gaussian and hence, in distribution, $||J_x u||$ is equal to a product of independent random variables:

**Proposition 6.2.** *For any $x \in \mathbb{R}^{n_0}$ and any unit vector $u \in \mathbb{R}^{n_0}$, the random variable $||J_x u||^2$ is equal in distribution to a product of independent scaled chi-squared random variables*

$$||J_x u||^2 \overset{d}{=} \frac{n_L}{n_0}\left(\prod_{\ell=1}^{L-1} \frac{2}{n_\ell}\chi^2_{\mathbf{n}_\ell}\right) \cdot \frac{1}{n_L}\chi^2_{n_L},$$

*where the number of degrees of freedom in the $\ell$th term of the product (for $\ell = 1, \ldots, L - 1$) is given by an independent binomial $\mathbf{n}_\ell \overset{d}{=} \text{Bin}(n_\ell, 1/2)$ with $n_\ell$ trials and success probability $1/2$. The number of degrees of freedom in the final term is deterministic and given by $n_L$.*

This Proposition has been derived a number of times in the literature (see Theorem 3 in Hanin (2018) and Fact 7.2 in Allen-Zhu et al. (2019)). We provide an alternative proof based on Proposition 2 in Hanin & Nica (2019) in the Supplementary Material. Note that the distribution of $||J_x u||$ is the same at every $x$ and $u$. Thus, fixing $x \in \mathbb{R}^{n_0}$ and a unit vector $u \in \mathbb{R}^{n_0}$, we find

$$\mathbb{E}\left[\text{len}(\mathcal{N}(M))^2\right] \leq \mathbb{E}\left[||J_x u||^2\right].$$

To prove (8), we note that $\mathbb{E}\left[\chi^2_{\mathbf{n}}\right] = n/2$ and apply Lemma 6.2 to find, as desired,

$$\mathbb{E}\left[||J_x u||^2\right] = \frac{n_L}{n_0}\left(\prod_{\ell=1}^{L-1}\mathbb{E}\left[\frac{2}{n_\ell}\chi^2_{\mathbf{n}_\ell}\right]\right)\mathbb{E}\left[n_L^{-1}\chi^2_{n_L}\right] = \frac{n_L}{n_0}\prod_{\ell=1}^{L-1}\left\{\frac{2}{n_\ell}\mathbb{E}\left[\mathbf{n}_\ell\right]\right\} = \frac{n_L}{n_0}. \quad \square$$

## 7 LIMITATIONS AND FUTURE WORK

We show that deep ReLU networks with appropriate initialization do not appreciably distort lengths and volumes, contrary to prior assumptions of exponential growth. Specifically, we provide an exact expression for the mean length distortion, which is bounded above by 1 and decreases slightly with increasing depth. We also prove that higher moments of the length distortion admit well-controlled upper bounds, and generalize this to distortion of higher dimensional volumes. We show empirically that our theoretical results closely match observations, unlike previous loose lower bounds.

There are several notable limitations of this paper, which offer promising directions for future work. First, we prove statements for networks at initialization, and our results do not necessarily hold after training; analyzing such behavior formally would require consideration of the loss function, optimizer, and data distribution. Second, our results, as stated, do not apply to convolutional, residual, or recurrent networks. We envision generalizations for networks with skip connections being straightforward, while formulations for convolutional and recurrent networks will likely be more complicated. In Appendix A, we provide preliminary results suggesting that expected length distortion decreases modestly with depth in both convolutional networks and those with skip connections. Third, we believe that various other measures of complexity for neural networks, such as curvature, likely demonstrate similar behavior to length and volume distortion in average-case scenarios with appropriate initialization. While somewhat parallel results for counting linear regions were shown in Hanin & Rolnick (2019; 2020), there remains much more to be understood for other notions of complexity. Finally, we look forward to work that explicitly ties properties such as low length distortion to improved generalization, as well as new learning algorithms that leverage this line of theory in order to better control inductive biases to fit real-world tasks.

## 8 ETHICS STATEMENT

This paper falls within deep learning theory and is intended to advance innovation for more effective and reliable algorithms. However, insofar as our work shows that deep ReLU networks at initialization are better-behaved than was previously believed, this could encourage the use of deeper neural networks. While deeper networks may enhance performance in some contexts, larger numbers of parameters are also associated with increased computational cost, which can both increase greenhouse gas emissions and exacerbate inequities caused by differential access to computational resources.

## 9 ACKNOWLEDGEMENTS

The authors gratefully acknowledge a range of funding sources that supported this research. BH would like to acknowledge NSF grants DMS-1855684 DMS-2133806 as well as an NSF CA-REER grant DMS-2143754 and an ONR MURI on Foundations of Deep Learning. DR would like to acknowledge support from the Natural Sciences and Engineering Research Council of Canada (NSERC) Discovery Grants program and the Canada CIFAR AI Chairs program.

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

## A  EXPERIMENTS FOR CONVOLUTIONAL AND RESIDUAL NETWORKS

We here provide preliminary experimental results for the mean length distortion in networks with convolutional layers and skip connections.

In Figure 4(a), we show results for networks with sequential convolutional layers (without pooling layers), initialized as before with weights i.i.d. normal with variance 2/fan-in, and a final fully connected layer. We use input dimension $n_0 = 16 \times 16 \times 3 = 768$ and output dimension 5. Our results indicate that the mean length distortion is, as expected, approximately equal to $\sqrt{n_L/n_0} \approx 0.08$, and that it decays slightly with depth.

In Figure 4(b), we consider residual networks. Here, the overall network $\mathcal{N}$ is defined in terms of residual modules $\mathcal{N}_1, \mathcal{N}_2, \ldots, \mathcal{N}_L$ and scales $\eta_1, \eta_2, \ldots, \eta_L$ according to:

$$\mathcal{N}(x) = x + \eta_1 \mathcal{N}_1(x) + \eta_2 \mathcal{N}_2(x + \eta_1 \mathcal{N}_1(x)) + \cdots$$
$$+ \eta_L \mathcal{N}_L \left( x + \eta_1 \mathcal{N}_1(x) + \eta_2 \mathcal{N}_2(x + \eta_1 \mathcal{N}_1(x)) + \cdots \right).$$

We set all residual modules $\mathcal{N}_\ell$ to be two-layer, fully connected ReLU networks. In keeping with Hanin & Rolnick (2018), we initialize all weights i.i.d. normal with variance 2/fan-in, while setting $\eta_1 = \eta_2 = \cdots = \eta_L = 1/L$. Our results suggest that mean length distortion again decays modestly overall, with a somewhat sharper decrease for small depths.

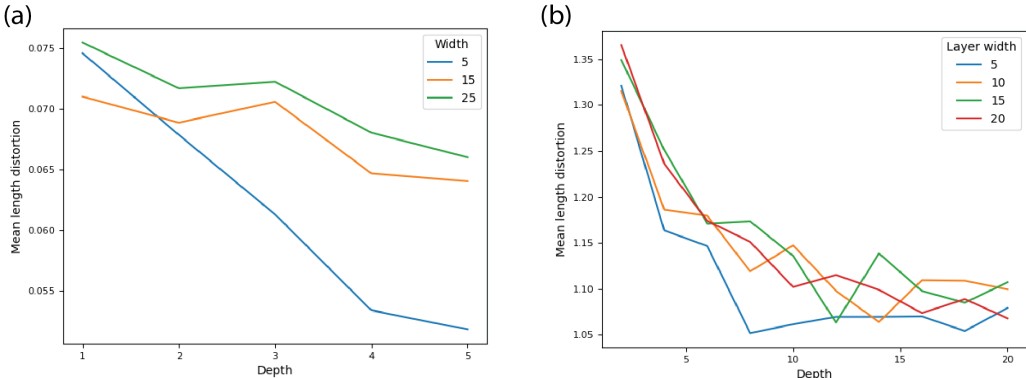

Figure 4: Mean length distortion as a function of depth, for networks with convolutional layers and skip connections, initialized using He normal initialization He et al. (2015). (a) shows results for networks having convolutional layers, where the input dimension $n_0$ equals $16 \times 16 \times 3 = 768$ and output dimension $n_L$ equals 5. Here width corresponds to the number of $3 \times 3$ kernels in each layer. As expected, the mean length distortion is $\sqrt{n_L/n_0} \approx 0.08$ and decays modestly with depth. (b) shows results for networks with skip connections occurring between even layers (i.e. each residual block is a fully-connected ReLU network of depth 2), again taking a line segment of unit length as the input curve. Here, depth corresponds to the total number of layers in the network, so that a depth-20 network includes 10 residual blocks. Both plots (a) and (b) show the empirical mean over 100 initializations of the weights and biases.

## B  EXPERIMENTAL DETAILS

For all experiments, weights were initialized from i.i.d. normal distributions with variance 2/fan-in and bias variance 0.1. We run several experiments that involve computing the length distortion of a given line segment in $\mathbb{R}^{n_0}$. We remark on how this was done, for which we rely on the notion of linear regions and bent hyperplanes associated with ReLU networks, explored in Hanin & Rolnick (2020); Telgarsky (2015). Specifically, ReLU networks partition input space into a collection of convex polytopes, which we call linear regions, as (generically) distinct linear functions are defined on each region corresponding to a different subset of the hidden neurons being activated (where a neuron is activated if its pre-activation is nonnegative). The boundaries of these linear regions are given by sets of points for which a particular neuron has preactivation equal to 0.

Say we are given a line segment $\ell$ with endpoints $\mathbf{p}_1$, $\mathbf{p}_2 \in \mathbb{R}^{n_0}$ parametrized by unit speed on $[0, 1]$, given by the set $\{\mathbf{p}_1 + t(\mathbf{p}_2 - \mathbf{p}_1) : t \in [0, 1]\}$, for which we aim to compute the length distortion. Let $\mathbf{w}^\ell = \mathbf{p}_2 - \mathbf{p}_1$.

We first approximate the intersections of the line segment with linear region boundaries using a binary search subroutine. Specifically, initialize the set $\mathcal{S} = [0, 0.5, 1]$, which will contain the parameter values for these approximations. For each iteration of the subroutine, we run the following procedure: for any three consecutive points $t_1, t_2, t_3 \in \mathcal{S}$, let $\mathbf{x}_1 = \mathbf{p}_1 + t_1(\mathbf{p}_2 - \mathbf{p}_1)$, $\mathbf{x}_2 = \mathbf{p}_1 + t_2(\mathbf{p}_2 - \mathbf{p}_1)$, $\mathbf{x}_3 = \mathbf{p}_1 + t_3(\mathbf{p}_2 - \mathbf{p}_1)$, and consider the values $\frac{||\mathcal{N}(\mathbf{x}_2) - \mathcal{N}(\mathbf{x}_1)||}{||\mathbf{x}_2 - \mathbf{x}_1||}$ and $\frac{||\mathcal{N}(\mathbf{x}_3) - \mathcal{N}(\mathbf{x}_2)||}{||\mathbf{x}_3 - \mathbf{x}_2||}$. These values are equal if the three points are in the same linear region, in which case we eliminate $t_1$ and $t_3$ from $\mathcal{S}$. If not equal, then there exists a linear region boundary in the segment from $\mathbf{x}_1$ to $\mathbf{x}_3$; we add the points $\frac{t_1 + t_2}{2}$ and $\frac{t_2 + t_3}{2}$ to $\mathcal{S}$. We iterate this procedure (15 times in our implementation); for the final iteration, we ensure that only the last point associated with a given linear region is in $\mathcal{S}$.

Now take the set $\mathcal{S} = [t_1, t_2, \ldots, t_n]$ returned by the binary search procedure; we proceed to compute the parameter values denoting the exact intersections of the line segment with region boundaries, which we shall store in $\mathcal{S}^*$. For consecutive points $t_i, t_{i+1} \in \mathcal{S}$, $i = 1, \ldots, n - 1$, determine the set of activated neurons for both points at $\mathbf{x}_i = \mathbf{p}_1 + t_i(\mathbf{p}_2 - \mathbf{p}_1)$, $\mathbf{x}_{i+1} = \mathbf{p}_1 + t_{i+1}(\mathbf{p}_2 - \mathbf{p}_1)$, and find the neuron at which these sets differ; we solve a linear equation to determine the value $t^* \in [0, 1]$ for which this neuron is 0, and replace $t_i$ with the exact value $t_i^*$ in $\mathcal{S}^*$. Finally, we append 0 and 1 to the respective ends of $\mathcal{S}^* = [t_0^* = 0, t_1^*, \ldots, t_n^* = 1]$.

Computing the length distortion is reduced to summing the lengths of the output segments corresponding to consecutive pairs $t_i, t_{i+1} \in \mathcal{S}^*$. Namely, for each such pair, the network is given by a single linear function on the segment of inputs between $\mathbf{x}_i = \mathbf{p}_1 + t_i(\mathbf{p}_2 - \mathbf{p}_1)$ and $\mathbf{x}_{i+1} = \mathbf{p}_1 + t_{i+1}(\mathbf{p}_2 - \mathbf{p}_1)$. We calculate the weight matrix $\mathbf{W}_i$ of this linear function – the length of the corresponding output segment is the product of $\mathbf{W}_i$ with the vector $\mathbf{w}^\ell$. The total length distortion is then given by the sum

$$\sum_{i=0}^{n-1} ||\mathbf{W}_i \mathbf{w}^\ell|| (t_{i+1}^* - t_i^*)$$

## C  PROOF OF THEOREM 5.1

Our first step in proving Theorem 5.1 is to obtain bounds on the moments of $\mathrm{len}(\mathcal{N}(M))$ in terms of the input-output Jacobian of $\mathcal{N}$ at a single point, which we recall was defined in (10). To accomplish them, we recall the notation from §6. Namely, fix a smooth unit speed parameterization of $M = \gamma([0, 1])$ with

$$\gamma : \mathbb{R} \to \mathbb{R}^{n_0}, \qquad \gamma(t) = (\gamma_1(t), \ldots, \gamma_{n_0}(t)).$$

The mapping

$$\Gamma := \mathcal{N} \circ \gamma, \qquad \Gamma : \mathbb{R} \to \mathbb{R}^{n_L}$$

gives a parameterization of the curve $\mathcal{N}(M)$, and we have

$$\mathrm{len}(\mathcal{N}(M)) = \int_0^1 ||\Gamma'(t)|| \, dt. \tag{14}$$

Let us note an important but ultimately benign technicality. The Jacobian $J_x$ of the map $x \mapsto \mathcal{N}(x)$ is not defined at every $x$ (namely those $x$ where some neuron turns from on to off). Thus, *a priori*, $\Gamma'(t)$ exists only at those $t$ for which $J_{\gamma(t)}$ exists. However, the formula (14) is still valid. Indeed, for any setting of weights and biases of $\mathcal{N}$ the map $\Gamma$ is Lipschitz. Thus, by Rademacher's theorem, $\Gamma'(t)$ exists for almost every $t \in [0, 1]$ and the length of the curve is given by integrating the norm of this almost-everywhere defined derivative. The following simple Lemma is a generalization of Lemma 6.1 and allows us to bound all moments of $\mathrm{len}(\mathcal{N}(M))$ in terms of the moments of the norm of the Jacobian vector product $||J_x u||$.

**Lemma C.1.** *For any integer $m \geq 1$, we have*

$$\mathbb{E}\left[\mathrm{len}(\mathcal{N}(M))^m\right] \leq \int_0^1 \mathbb{E}\left[\left|\left|J_{\gamma(t)} \gamma'(t)\right|\right|^m\right] dt. \tag{15}$$

*Proof.* Taking powers in (14) and using Tonelli's theorem to interchange the expectation and integrals, we obtain

$$\mathbb{E}\left[\operatorname{len}(\mathcal{N}(M))^m\right] = \int_0^1 \cdots \int_0^1 \mathbb{E}\left[\prod_{j=1}^m ||\Gamma'(t_j)||\right] dt_1 \cdots dt_m. \tag{16}$$

To proceed, let us recall a special case of the power mean inequality which says that for any $a_i \geq 0$ we have

$$\prod_{i=1}^m a_i \leq \frac{1}{m} \sum_{i=1}^m a_i^m$$

Applying this to the integrand in (16), we conclude

$$\mathbb{E}\left[\operatorname{len}(\mathcal{N}(M))^m\right] \leq \int_0^1 \mathbb{E}\left[||\Gamma'(t)||^m\right] dt. \tag{17}$$

Next, fix $t \in [0,1]$. For any neuron $z$ in $\mathcal{N}$, denote by $x \mapsto z(x)$ its pre-activation. Note that $\mathbb{P}(z(\gamma(t)) = 0) = 0$ since our bias variance $C_b$ is set to some fixed positive constant and hence the bias of each neuron has a continuous density. Therefore, with probability 1 over the randomness in the weights and biases of $\mathcal{N}$, there exists a neighborhood $\mathcal{U}$ of $\gamma(t)$ on which $z(x)$ has constant sign for all $x \in \mathcal{U}$. The Jacobian $J_{\gamma(t)}$ is therefore well-defined and the chain rule yields

$$\Gamma'(t) = J_{\gamma(t)}\gamma'(t). \tag{18}$$

Substituting this into (17) completes the proof. $\square$

Having obtained the expression (15), we have reduced bounding the moments of $\operatorname{len}(\mathcal{N}(M))$ to bounding the moments of the random variable $||J_x u||$ for a fixed $x \in \mathbb{R}^{n_0}$ and unit vector $u \in \mathbb{R}^{n_0}$. Prior work (e.g. Thm. 1 in Hanin & Nica (2019)) shows that these moments satisfy

$$\mathbb{E}\left[||J_x u||^{2m}\right] = \left(\frac{n_L}{n_0}\right)^m \exp\left(5\binom{m}{2}\sum_{\ell=1}^{L-1} n_\ell^{-1} + O\left(\sum_{\ell=1}^{L-1} n_\ell^{-2}\right)\right),$$

provided

$$\binom{m}{2} < \min_{\ell=1,\ldots,L-1} n_\ell. \tag{19}$$

Substituting these moment estimates in (15) completes the derivation of (4). However, the results in Hanin & Nica (2019) are subtle because they apply to any distribution of weights and biases. They also give the slightly sub-optimal restriction (19) that $m^2$ must be smaller than a constant times the minimum of the $n_\ell$'s. In the special case where the distribution of weights and biases is Gaussian, we can do slightly better by computing the moments of $||J_x u||$ more directly (note that in the statement of Theorem 5.1, we required only that $m$ is smaller than a constant times the minimum of the $n_\ell$'s). We will need this anyway to derive the slightly more refined estimates in (2) and (3). Let us therefore check that, in distribution, $||J_x u||$ is equal to a product of independent random variables:

**Proposition C.2.** *For any $x \in \mathbb{R}^{n_0}$ and any unit vector $u \in \mathbb{R}^{n_0}$, the random variable $||J_x u||^2$ is equal in distribution to a product of independent scaled chi-squared random variables*

$$||J_x u||^2 \overset{d}{=} \frac{n_L}{n_0}\left(\prod_{\ell=1}^{L-1} \frac{2}{n_\ell}\chi^2_{\mathbf{n}_\ell}\right) \cdot \frac{1}{n_L}\chi^2_{n_L},$$

*where the number of degrees of freedom in the $\ell$th term of the product (for $\ell = 1, \ldots, L-1$) is given by an independent binomial*

$$\mathbf{n}_\ell \overset{d}{=} \operatorname{Bin}(n_\ell, 1/2)$$

*with $n_\ell$ trials and success probability $1/2$. The number of degrees of freedom in the final term is deterministic and given by $n_L$.*

*Proof.* Consider a ReLU network $\mathcal{N}$ with input dimension 1, output dimension $n_L$, and hidden layer widths $n_1, \ldots, n_{L-1}$. Suppose the weight matrices $W^{(\ell)}$ and bias vectors $b^{(\ell)}$ are independent with i.i.d. Gaussian components:

$$W_{ij}^{(\ell)} \sim G(0, 2/n_{\ell-1}), \qquad b_j^{(\ell)} \sim G(0, C_b),$$

where $C_b > 0$ is any fixed constant. For a fixed network input $x \in \mathbb{R}^{n_0}$, with probability 1, the input-output Jacobian $J_x$ is well-defined. Moreover, it can be written as

$$J_x = W^{(L)} D^{(L-1)} W^{(L-1)} \cdots D^{(1)} W^{(1)},$$

where $W^{(\ell)}$ is the matrix of weights from layer $\ell - 1$ to layer $\ell$ and $D^{(\ell)}$ is an $n_\ell \times n_\ell$ diagonal matrix:

$$D^{(\ell)} = \mathrm{Diag}\left(\mathbf{1}_{\left\{z_i^{(\ell)} \geq 0\right\}}, i = 1, \ldots, n_\ell\right)$$

whose diagonal entries are 0 or 1 depending on whether the pre-activation $z_i^{(\ell)}$ of neuron $i$ in layer $\ell$ is positive at our fixed input $x$. Next, according to Proposition 2 in Hanin & Nica (2019), the marginal distribution of each $D^{(\ell)}$ is that its diagonal entries are independent Bernoulli$(1/2)$ random variables. Moreover, we have the following equality in distribution:

$$J_x \stackrel{d}{=} \eta W^{(L)} \mathcal{D}^{(L-1)} W^{(L-1)} \cdots \mathcal{D}^{(1)} W^{(1)}$$

where $\mathcal{D}^{(\ell)}$ are independent of each other (and of $W^{(\ell)}$) resampled from the marginal distribution of $D^{(\ell)}$ and $\eta$ is an independent diagonal matrix with independent diagonal entries that are $\pm 1$ with equal probability. In particular, for a fixed unit vector $u \in \mathbb{R}^{n_0}$, we have

$$\|J_x u\| \stackrel{d}{=} \|W^{(L)} \mathcal{D}^{(L-1)} W^{(L-1)} \cdots \mathcal{D}^{(1)} W^{(1)} u\|.$$

We may rewrite this as

$$\|W^{(L)} \mathcal{D}^{(L-1)} W^{(L-1)} \cdots \mathcal{D}^{(2)} W^{(2)} u^{(1)}\| \, \|\mathcal{D}^{(1)} W^{(1)} u\|, \tag{20}$$

for $u^{(1)} := \frac{\mathcal{D}^{(1)} W^{(1)} u}{\|\mathcal{D}^{(1)} W^{(1)} u\|}$, where we interpret the expression (20) as equal to 0 if $\mathcal{D}^{(1)}$ is the zero matrix. To complete the proof, we need the following standard observation.

**Lemma C.3.** *Suppose $W$ is an $n \times n'$ matrix with i.i.d. Gaussian entries and $u$ is a random unit vector in $\mathbb{R}^{n'}$ that is independent of $W$ but otherwise has any distribution. Then $Wu$ is independent of $u$ and is equal in distribution to $Wv$ where $v$ is any fixed unit vector in $\mathbb{R}^{n'}$.*

*Proof.* For any *fixed* orthogonal matrix $\mathcal{O}$, it is a standard fact that $W\mathcal{O}$ is equal in distribution to $W$. Thus, for any measurable sets $A \subseteq \mathbb{R}^n$ and $B \subseteq \mathbb{R}^{n'}$, since $u, W$ are independent we have

$$\mathbb{P}(Wu \in A, u \in B) = \int_{S^{n'-1}} \mathbb{P}(Wu_0 \in A) \, d\mathbb{P}_u(u_0),$$

where $\mathbb{P}_u(u_0)$ is the distribution of $u$. Fix any $u_0 \in S^{n'-1}$ and let $\mathcal{O} = \mathcal{O}(u_0)$ be any orthogonal matrix so that $u_0 = \mathcal{O} e_1$ with $e_1 = (1, 0, \ldots, 0)$ is the first standard unit vector. Then, since $W\mathcal{O}$ is equal in distribution to $W$, we have

$$\mathbb{P}(Wu_0 \in A) = \mathbb{P}(We_1 \in A),$$

which is independent of $u_0$. We therefore find

$$\mathbb{P}(Wu \in A, u \in B) = \mathbb{P}(We_1 \in A) \, \mathbb{P}(u \in B),$$

as desired. $\qquad\square$

We are now in a position to complete the proof of Proposition C.2. Combining Lemma C.3 with (20), we find that, in distribution $\|J_x u\|$ equals

$$\|W^{(L)} \mathcal{D}^{(L-1)} W^{(L-1)} \cdots \mathcal{D}^{(2)} W^{(2)} u\| \, \|\mathcal{D}^{(1)} W^{(1)} u\|. \tag{21}$$

Note that these two terms are independent. Repeating this argument, we obtain that $||J_x u||$ is equal in distribution to the product

$$||W^{(L)} u|| \, ||\mathcal{D}^{(L-1)} W^{(L-1)} u|| \cdots ||\mathcal{D}^{(1)} W^{(1)} u||. \tag{22}$$

The terms in this product are independent. To complete the proof note that for $\ell = 1, \ldots, L-1$ the number of non-zero entries in $\mathcal{D}^{(\ell)}$ is a binomial random variable $\mathbf{n}_\ell$ with $n_\ell$ trials, each with probability of success $1/2$ and that

$$||\mathcal{D}^{(\ell)} W^{(\ell)} u||^2$$

is precisely $2/n_\ell$ times a sum of $\mathbf{n}_\ell$ squares of independent standard Gaussians. Thus, for $\ell = 1, \ldots, L-1$,

$$||\mathcal{D}^{(\ell)} W^{(\ell)} u||^2 \stackrel{d}{=} \frac{2}{n_{\ell-1}} \chi^2_{\mathbf{n}_\ell}. \tag{23}$$

Similarly

$$||W^{(L)} u||^2 \stackrel{d}{=} \frac{2}{n_{L-1}} \chi^2_{n_L}. \tag{24}$$

Substituting (23) and (24) into (22) completes the proof. $\square$

Evaluating the moments of $\mathrm{len}(\mathcal{N}(M))$ is now a matter of computing the moments of some scaled chi-squared random variables with a random number of degrees of freedom. For instance, recalling that

$$\mathbb{E}\left[\chi^2_k\right] = k$$

and applying Lemma C.2 as well as the tower property of the expectation we find

$$\mathbb{E}\left[||J_x u||^2\right] = \frac{n_L}{n_0} \left( \prod_{\ell=1}^{L-1} \mathbb{E}\left[\frac{2}{n_\ell} \chi^2_{\mathbf{n}_\ell}\right] \right) \mathbb{E}\left[n_L^{-1} \chi^2_{n_L}\right]$$

$$= \frac{n_L}{n_0} \prod_{\ell=1}^{L-1} \left\{ \frac{2}{n_\ell} \mathbb{E}\left[\mathbb{E}\left[\chi^2_{\mathbf{n}_\ell} \mid \mathbf{n}_\ell\right]\right] \right\}$$

$$= \frac{n_L}{n_0} \prod_{\ell=1}^{L-1} \left\{ \frac{2}{n_\ell} \mathbb{E}\left[\mathbf{n}_\ell\right] \right\}$$

$$= \frac{n_L}{n_0}.$$

Substituting this into (15) with $m = 2$ yields

$$\mathrm{Var}\left[\mathrm{len}(\mathcal{N}(M))\right] \leq \frac{n_L}{n_0},$$

yielding (3). To prove (2), we need to estimate $\mathbb{E}\left[\mathrm{len}\,\mathcal{N}(M)\right]$. By taking expectations in (14) and using (18), we find

$$\mathbb{E}\left[\mathrm{len}\,\mathcal{N}(M)\right] = \int_0^1 \mathbb{E}\left[||J_{\gamma(t)} \gamma'(t)||\right] dt = \mathbb{E}\left[||J_x u||\right],$$

where we've used that by Lemma C.2, the distribution of $||J_x u||$ is the same for every $x \in \mathbb{R}^{n_0}$ and every unit vector $u$. Moreover, again using Lemma C.2, we see that $\mathbb{E}\left[||J_x u||\right]$ equals

$$\left(\frac{n_0}{n_L}\right)^{1/2} \prod_{\ell=1}^{L-1} \mathbb{E}\left[\left(\frac{2}{n_\ell} \chi^2_{\mathbf{n}_\ell}\right)^{1/2}\right] \mathbb{E}\left[\left(\frac{1}{n_L} \chi^2_{n_L}\right)^{1/2}\right].$$

To simplify this, a direct computation shows that

$$\mathbb{E}\left[\left(k^{-1} \chi^2_k\right)^{1/2}\right] = \frac{\Gamma\left(\frac{k+1}{2}\right)}{\Gamma\left(\frac{k}{2}\right) \left(\frac{k}{2}\right)^{1/2}},$$

where $\Gamma(\cdot)$ is the Gamma function. Next, for any positive random variable $X$ with $\mathbb{E}[X] = 1$ we have

$$\mathbb{E}\left[X^{1/2}\right] = \mathbb{E}\left[(1 + (X - 1))^{1/2}\right]$$
$$= 1 - \frac{1}{8}\text{Var}[X] + O(|X - 1|^3).$$

Recall that

$$\text{Var}[\chi_k^2] = 2k.$$

Using this and the law of total variance yields

$$\mathbb{E}\left[\left(\frac{2}{n_\ell}\chi_{\mathbf{n}_\ell}^2\right)^{1/2}\right] = 1 - \frac{1}{2n_\ell^2}\text{Var}[\chi_{\mathbf{n}_\ell}^2] + O(n_\ell^{-2})$$

$$= 1 - \frac{1}{2n_\ell^2}\left[\mathbb{E}\left[\text{Var}[\chi_{\mathbf{n}_\ell}^2 \mid \mathbf{n}_\ell]\right]\right.$$

$$\left. + \text{Var}[\mathbb{E}\left[\chi_{\mathbf{n}_\ell}^2 \mid \mathbf{n}_\ell]\right]\right] + O(n_\ell^{-2})$$

$$= 1 - \frac{1}{2n_\ell^2}\left[\mathbb{E}\left[2\mathbf{n}_\ell\right] + \text{Var}[\mathbf{n}_\ell]\right] + O(n_\ell^{-2})$$

$$= 1 - \frac{1}{2n_\ell^2}\left[n_\ell + \frac{n_\ell}{4}\right] + O(n_\ell^{-2})$$

$$= 1 - \frac{5}{8n_\ell} + O(n_\ell^{-2}).$$

Thus, we find that $\mathbb{E}\left[\text{len}(\mathcal{N}(M))\right]$ equals

$$\left(\frac{n_L}{n_0}\right)^{1/2}\frac{\Gamma\left(\frac{n_L+1}{2}\right)}{\Gamma\left(\frac{n_L}{2}\right)\left(\frac{n_L}{2}\right)^{1/2}} \times \exp\left[-\frac{5}{8}\sum_{\ell=1}^{L-1}n_\ell^{-1} + O\left(\sum_{\ell=1}^{L-1}n_\ell^{-2}\right)\right],$$

as claimed. Finally, let us check the higher moment estimates (4). By Lemma C.2 we have the following estimate

$$||J_x u||^2 = \frac{n_L}{n_0}\left(\prod_{\ell=1}^{L-1}\frac{2}{n_\ell}\chi_{\mathbf{n}_\ell}^2\right)n_L^{-1}\chi_{n_L}^2$$

$$\leq \frac{n_L}{n_0}\exp\left[\sum_{\ell=1}^{L-1}\left\{\frac{2}{n_\ell}\chi_{\mathbf{n}_\ell}^2 - 1\right\}\right]n_L^{-1}\chi_{n_L}^2,$$

where we used that $x = x - 1 + 1 \leq e^{x-1}$. For any $m$, we have

$$\mathbb{E}\left[\left(\frac{1}{n_L}\chi_{n_L}^2\right)^m\right] = \left(1 + \frac{2}{n_L}\right)\cdots\left(1 + \frac{2m-2}{n_L}\right)$$

$$\leq \exp\left[\sum_{j=0}^{m-1}\frac{2j}{n_L}\right] \leq \exp\left[\frac{m^2}{n_L}\right].$$

Therefore, $||J_x u||^{2m}$ is bounded above by

$$\left(\frac{n_L}{n_0}\right)^m\exp\left[m\sum_{\ell=1}^{L-1}\left\{\frac{2}{n_\ell}\chi_{\mathbf{n}_\ell}^2 - 1\right\} + \frac{m^2}{n_L}\right]. \tag{25}$$

Finally, for any fixed positive integer $n$ we may write

$$\frac{2}{n}\chi_{\mathbf{n}}^2 - 1 = \frac{2}{n}\sum_{k=1}^{n}\left\{\xi_k Z_k^2 - \frac{1}{2}\right\},$$

where $\xi_k$ are independent Bernoulli$(1/2)$ random variables and $Z_k$ are independent standard Gaussians. A direct computation shows that the centered random variables $\xi_k Z_k^2 - \frac{1}{2}$ are sub-exponential $\text{SE}(\nu^2, \alpha)$ with some universal parameters $\nu^2, \alpha$. Thus, by the stability of sub-exponential random variables under summation we have

$$\frac{2}{n_\ell} \chi^2_{\mathbf{n}_\ell} - 1 \in \text{SE}\left(\frac{4\nu^2}{n_\ell}, \frac{2\alpha}{n_\ell}\right).$$

Again using this property we conclude

$$\sum_{\ell=1}^{L-1} \frac{2}{n_\ell} \chi^2_{\mathbf{n}_\ell} - 1 \in \text{SE}\left(4\nu^2 \sum_{\ell=1}^{L-1} \frac{1}{n_\ell}, \frac{2\alpha}{n_*}\right),$$

where

$$n_* = \min\{n_1, \ldots, n_{L-1}\}.$$

Therefore, by (25), we find that

$$\mathbb{E}\left[||J_x u||^{2m}\right] \leq \left(\frac{n_L}{n_0}\right)^m \mathbb{E}\left[e^{mY}\right] \exp\left[\frac{m^2}{n_L}\right],$$

where $Y \in \text{SE}\left(4\nu^2 \sum_{\ell=1}^{L-1} \frac{1}{n_\ell}, \frac{2\alpha}{n_*}\right)$. By definition of sub-exponential random variables, we have

$$\mathbb{E}\left[e^{mY}\right] \leq \exp\left[4m^2\nu^2 \sum_{\ell=1}^{L-1} \frac{1}{n_\ell}\right],$$

provided $m < n_*/2\alpha$ for some universal constant $c > 0$. This completes the derivation of (4) and completes the proof of Theorem 5.1. $\qquad\square$

## D  PROOF OF THEOREM 5.2

The proof of Theorem 5.2 follows closely the argument for Theorem 5.1. For a given input $x \in \mathbb{R}^{n_0}$, we will continue to write

$$J_{\mathcal{N}}(x) := \left(\partial_{x_j} \mathcal{N}_i(x)\right)_{\left\{\substack{1 \leq j \leq n_0 \\ 1 \leq i \leq n_L}\right\}}$$

for the input-output Jacobian of the network function, which exists for Lebesgue almost every $x \in \mathbb{R}^{n_0}$. We will write $\Pi_{T_x M} : \mathbb{R}^{n_0} \to T_x M$ for the orthogonal projection onto this tangent space. We have

$$\text{vol}_d(\mathcal{N}(M)) = \int_M \left(\det\left(\Pi_{T_x M} J_{\mathcal{N}}(x)^T J_{\mathcal{N}}(x) \Pi_{T_x M}\right)\right)^{1/2} \text{vol}_d(dx),$$

where $\text{vol}_d$ is the $d-$dimensional Hausdorff measure. Indeed, the integrand measures, at each $x \in M$, the volume of the image of an infinitesimal cube on $M$ of volume $\text{vol}_d(dx)$ centered at $x$ under the map $x \mapsto \mathcal{N}(x)$. Arguing precisely as in the proof of Lemma C.1, we find that for any integer $m \geq 1$

$$\mathbb{E}\left[\text{vol}_d(\mathcal{N}(M))^m\right] \leq \int_M \mathbb{E}\left[\det\left(\Pi_{T_x M} J_{\mathcal{N}}(x)^T J_{\mathcal{N}}(x) \Pi_{T_x M}\right)\right]^{m/2} \text{vol}_d(dx) \qquad (26)$$

Fix $x \in M$ and denote by

$$e_1(x), \ldots, e_d(x)$$

an orthonormal basis of the tangent space of $M$. Then, by the Gram identity, we may write

$$\det\left(\Pi_{T_x M} J_{\mathcal{N}}(x)^T J_{\mathcal{N}}(x) \Pi_{T_x M}\right) = ||J_{\mathcal{N}}(x)e_1(x) \wedge \cdots \wedge J_{\mathcal{N}}(x)e_d(x)||^2, \qquad (27)$$

where we recall that the wedge product is the anti-symmetrization of the tensor product. Just as in the proof of Theorem 5.1, the key observation is that for Gaussian weights we have that $||J_{\mathcal{N}}(x)e_1(x) \wedge \cdots \wedge J_{\mathcal{N}}(x)e_d(x)||$ is a product of i.i.d. random variables. The formal statement is the following:

**Proposition D.1.** *For any $x \in \mathbb{R}^{n_0}$ and collection of orthonormal unit vectors $u_1, \ldots, u_d \in \mathbb{R}^{n_0}$, the random variable*

$$||J_x u_1 \wedge \cdots \wedge J_x u_d||^2$$

*is equal in distribution to a product of independent scaled chi-squared random variables:*

$$\left(\frac{n_L}{n_0}\right)^d \left(\prod_{\ell=1}^{L-1} \prod_{j=1}^d \frac{2}{n_\ell} \chi^2_{\mathbf{n}_\ell - j + 1}\right) \cdot \prod_{j=1}^d \frac{1}{n_L} \chi^2_{\mathbf{n_L} - j + 1},$$

*where all the terms in the product are independent and for $\ell = 1, \ldots, L-1$ we've written $\mathbf{n}_\ell$ for independent Binomial random variables:*

$$\mathbf{n}_\ell \overset{d}{=} \mathrm{Bin}\left(n_\ell, 1/2\right)$$

*with $n_\ell$ trials and success probability $1/2$.*

*Proof.* The proof is identical to that of Lemma C.2. The only difference is that we must invoke the following amplification of Lemma C.3:

**Lemma D.2.** *Let $u_1, \ldots, u_k \in \mathbb{R}^{n'}$ be a collection of orthonormal vectors (i.e. a collection) with any distribution. Let $W$ be an independent $n \times n'$ matrix with i.i.d. Gaussian entries. Then*

$$W(u_1 \wedge \cdots \wedge u_k) = W u_1 \wedge \cdots \wedge W u_k$$

*is independent of $u_1 \wedge \cdots \wedge u_k$ and is equal in distribution to $W v_1 \wedge \cdots W v_k$ where $v_1, \ldots, v_k$ is any fixed collection of orthonormal vectors.*

*Proof.* The proof is identical to that of Lemma C.3, except that we note that for that, given $u_1, \ldots, u_k$, there exists an orthogonal matrix $\mathcal{O} = \mathcal{O}(u_1, \ldots, u_k)$ so that

$$u_j = \mathcal{O} e_j, \qquad j = 1, \ldots, k$$

where $e_j$ are the standard basis vectors. $\qquad\square$

$\qquad\square$

With Lemma D.1 in hand, we complete the proof of Theorem 5.2 as follows. First, note that for any random variable $X$ we have

$$\mathbb{E}[X] \le \mathbb{E}[X^2]^{1/2}, \qquad \mathrm{Var}[X] \le \mathbb{E}[X^2].$$

Hence, Theorem 5.2 will follow once we show that

$$\mathbb{E}\left[\mathrm{vol}_d(\mathcal{N}(M))^2\right] \le \left(\frac{n_L}{n_0}\right)^d \exp\left[-\binom{d}{2}\sum_{\ell=1}^L n_\ell^{-1}\right].$$

Combining Proposition D.1 with (26) and (27), this estimate follows by showing that

$$\mathbb{E}\left[||J_x u_1 \wedge \cdots \wedge J_x u_d||^2\right] \le \left(\frac{n_L}{n_0}\right)^d \exp\left[-\binom{d}{2}\sum_{\ell=1}^L n_\ell^{-1}\right]. \qquad (28)$$

To check this, recall that

$$\mathbb{E}\left[\chi_k^2\right] = k.$$

Hence,

$$\mathbb{E}\left[||J_{\mathcal{N}}(x)e_1(x) \wedge \cdots \wedge J_{\mathcal{N}}(x)e_d(x)||^2\right] = \left(\frac{n_L}{n_0}\right)^d \left(\prod_{\ell=1}^{L-1} \prod_{j=1}^d \frac{2}{n_\ell} \mathbb{E}\left[\chi^2_{\mathbf{n}_\ell - j + 1}\right]\right) \cdot \prod_{j=1}^d \frac{1}{n_L} \mathbb{E}\left[\chi^2_{\mathbf{n_L} - j + 1}\right]$$

$$= \left(\frac{n_L}{n_0}\right)^d \prod_{\ell=1}^L \prod_{j=1}^d \left(1 - \frac{j-1}{n_\ell}\right)$$

$$\le \left(\frac{n_L}{n_0}\right)^d \exp\left[-\binom{d}{2}\sum_{\ell=1}^L n_\ell^{-1}\right],$$

where in the last line we used that $1 + x \le e^x$. This completes the proof. $\qquad\square$

