# OpenReview forum: "Deep ReLU Networks Preserve Expected Length"
_ICLR.cc/2022/Conference — ICLR 2022 Poster_

### Official Review · Reviewer_TwSU · 2021-10-22

**Correctness:** 4
**Technical Novelty And Significance:** 4
**Empirical Novelty And Significance:** 2
**Recommendation:** 8
**Confidence:** 4

**Main Review:**

This paper is well written and addresses an important problem using very careful derivations. It is a welcome addition to the literature, since it was not obvious that typical network initialization yields a non-explosion of length distortion as the network grows deeper.

The main limitations I see are: (1) the results apply to randomly initialized networks and not to trained networks; (2) the results apply to fully connected networks and not to architectures like ResNet or Transformer. The authors are well aware of these limitations, and preliminary empirical results for (2) are present in Appendix A. I don't believe any further improvement to address these limitations is needed.

The following are minor comments that I have.

* Page 3, third line of the first paragraph: “towards” → “that”
* Caption of Figure 1: can you increase the margins a bit? Otherwise the caption can be confused with the main text. Also, this figure is in the middle of the page, but it could be placed at the top or bottom of the page instead to avoid interruption (since it is not even referenced in the current section, so there is no use in interrupting the flow of text there).
* Page 4: it is quite confusing to have a discussion on Figure 1 side by side with Figure 2. Similarly, Figure 3 is far from the place where it is actually referenced. I think it would be good to re-organize the placement of all figures.
* End of Section 6: better to place the qed symbol on the same line as the equation (you can use \qedhere if you are inside a proof environment). This will also save you some space.

**Summary Of The Paper:**

This paper computes/estimates the mean and higher moments of the length distortion of fully connected ReLU networks at initialization, assuming typical random initialization of the weights. In particular, it shows that length distortion does not grow with the depth of the network (as previously believed). The paper also presents analog results for the average distortion of higher-dimensional volumes.

**Summary Of The Review:**

This paper provides a novel set of theorems on distortion of length and volume. It is well written and precise.

---

> ### Author Response · Authors · 2021-11-21
> **Detailed Response to Reviewer Comments**
>
> # Response
>
> We thank the reviewer for their careful reading, suggestions, and comments. We reply to them below in the same order as they are listed in the review.
> ## Comments
> * We have fixed this typo.
> * Indeed, this was potentially confusing. We have increased the space after the caption.
> * Thank you for this suggestion. We have moved Fig. 1 so it is next to Fig. 2 - and it indeed is much clearer. For Fig. 3, we explored various alternative placements but due to the column format and the many full-line equations in later sections, unfortunately we could not find another place that worked better.
> * We have moved the QED symbol as suggested.

---

### Official Review · Reviewer_MMSS · 2021-10-31

**Correctness:** 4
**Technical Novelty And Significance:** 3
**Empirical Novelty And Significance:** Not applicable
**Recommendation:** 6
**Confidence:** 3

**Main Review:**

Overall, this is a well-written paper that follows all the best practices of making a theoretical contribution easy to assimilate in deep learning: they carefully introduce the main topic of interest, describe their results in plain English, informally state their main results and the intuition behind them while also generously illustrating those results through figures, then state the precise results with proofs in the appendix, and wrap up with a clear definition of what was achieved and what is left as future work.

The authors even include a section in the appendix on CNNs to avoid the typical reviewer question about having tried it in CNNs: I feel your pain!

# Specific comments

In Section 2, when you say that "it has been shown that it is possible to set the weights of a deep ReLU network such that the number of linear regions computed by the network grows exponentially in the depth", you should also mention negative results in depth for linear regions such as the depth-width tradeoff discussed in [1].

When discussing expected results on linear regions, perhaps it would be worth also mentioning more recent results concerning maxout networks [2].

In Section 3, please rewrite this: "This phenomenon indicates an implicit regularization towards causes these learned functions to be surprisingly simple."

In Figure 1: What is the number of hidden layers used? What happens when it varies?

When you describe the setting in Section 3, it is not said explicitly that $L$ is the number of hidden layers. Hence, can you please rewrite "and hidden layer widths" as something like "and $L$ hidden layers with widths"?

Other small issues in Section 3:
- "Our primary object of the study": remove "the", or replace "of" with "in"
- "apply an $\epsilon$-net argument": please describe it in more detail

When I first skimmed the paper, I was disappointed that there was no conclusion, only to find out later than the first paragraph of Section 7 is indeed a conclusion. Perhaps it would be worth renaming Secton 7 as something like "Conclusion, Limitations, and Future Work".

In Section 7, I would rewrite "There are several notable limitations of this paper" as "This paper has several notable limitations".

As you discuss looking into more detail on convolutional networks, perhaps it would be worth citing related work on linear regions for CNNs [3].

# References cited

[1] https://arxiv.org/abs/1711.02114

[2] https://arxiv.org/abs/2107.00379

[3] https://arxiv.org/abs/2006.00978

**Summary Of The Paper:**

This paper presents results on the expected trajectory length of the outputs of a neural network at initialization with respect to the trajectory length of the inputs. The study of trajectory length is a spin-off of the study of linear regions in neural networks, since both are discussed in the first paper by Raghu et al. (2017). Notably, this paper analyzes trajectory length through the same lenses used in subsequent work on linear regions by Hanin and Rolnick (2019), which has shown that the number of linear regions for weights typically used at initialization does not allow the corresponding metric to grow exponentially on the depth of the network.

In addition, the authors point out at a missing coefficient in prior results on trajectory length that lead to very different conclusions; and they also provide results on higher moments of trajectory length as well as on the multi-dimensional case of expected volume.

**Summary Of The Review:**

The paper is well organized, adds more nuance to our understanding of the expressiveness of neural networks, but also leaves a taste of the expected length after training being an obvious follow-up paper which could have fit here as well. I would have been happier to see the whole affair discussed as a single and stronger paper, but I believe that the contributions here are enough to cross the acceptance line.

---

> ### Author Response · Authors · 2021-11-21
> **Detailed Response to Reviewer**
>
> # Response
>
> We thank the reviewer for their careful reading, suggestions, and comments. We reply to them below in the same order as they are listed in the review.
> ## Specific Comments
> * Thank you for pointing us to several additional references. We've added pointers to [1]-[3] in the related work section.
> * We changed the wording in "This phenomenon... " in section 3.
> * In Figure 1, the number of layers is indicated on the x-axis as the depth. In our normalization Depth $L$ means $L$ hidden layers (and so $L+2$ layers counting the network input layer and the affine output layer).
> * We've amended the start of Section 3 to explicitly state ``and $L$ hidden layers"
> * We have corrected the typo in ``our primary object of study...''
> * The $\epsilon$-net argument we have in mind goes like this. To bound from above the supremum $\sup_{x\in M} \ || J_xu || $ one can first fix $\epsilon>0$ and consider an $\epsilon$-net $N_\epsilon\subseteq M$, which is a finite collection of points for which every point in $M$ is not more than $\epsilon$ away from one point in the collection $N_\epsilon:$
>     $$
>     \forall x\in M~~~ \exists x'\in N_\epsilon\text{ s.t. } ||x-x'||\leq \epsilon.
>     $$
>     If we knew that $x\mapsto || J_xu || $ is $L-$Lipschitz, then we have
>     $$
>     \sup_{x\in K} || J_xu ||\leq L\epsilon + \sup_{x\in N_\epsilon} || J_xu ||
>     $$
>     A union bound then yields
>     $$
>     \mathbb P\left( \sup_{x\in N_\epsilon} || J_xu || > t\right)\leq |N_\epsilon| \mathbb P\left(|| J_xu || >t\right).
>     $$
>     Hence,
>     $$
>      \mathbb P \left(\sup_{x\in M} || J_xu || >L\epsilon+t\right)\leq |N_\epsilon| \mathbb P \left( || J_xu || >t\right).
>     $$
>      Applying the old results of Allen-Zhu et~al.~(or Hanin-Nica or a slew of other similar observations) gives sharp bounds on $\mathbb P \left( || J_xu || >t\right)$ and hence gives bounds on probability that the supremum of $|| J_xu || $ is large. However, and this is the key point, we are not aware of any estimates on the Lipschitz constant of the map $x\mapsto || J_xu || $ and hence cannot use this approach. Since there was not enough room to include the preceding explanation in the body of the article we've inserted instead a canonical reference to arguments in Vershynin's book on High Dimensional Probability that use $\epsilon$-nets to obtain concentration bounds along the lines we've indicated.
>
> * We completely agree regarding the title of the last section and have renamed it simply ``Conclusion".

---

> > ### Comment · Reviewer_MMSS · 2021-11-22
> > **Reviewer response**
> >
> > I appreciate the clarifications and effort by the authors to improve the paper based on the feedback.

---

### Official Review · Reviewer_mxm7 · 2021-11-02

**Correctness:** 4
**Technical Novelty And Significance:** 2
**Empirical Novelty And Significance:** 3
**Recommendation:** 6
**Confidence:** 4

**Main Review:**

The problem is interesting and the authors have motivated it well; However the results are far from surprising or novel.

The technical challenge is to bound the spectral norm of the Jacobian of each layer (for random weights). and then use independence.
Nevertheless, the appearance of ReLUs (non-linearities) makes things challenging. The main idea is effectively to prove that the Jacobian can be ``viewed" as a product of i.i.d Gaussian and Diagonal matrices with Bernoulli random variables on the diagonal.
These technical ideas have appeared in other papers before (e.g., Allen Zhu et al "A convergence theory for deep learning via overparameterization.").  What seems quite frustrating is that the authors mention in the abstract "it is widely believed that length grows exponentially in network depth". They authors prove the contrary, but this counterintuitive fact was already proved to some extent and therefore the claim in the abstract is misleading. Moreover, I would like to note that even the result about distortion on manifolds may seem novel, it is basically reduced to the 1-dimensional case in a straightforward manner.

Overall, this is some interesting but technically incremental work that I would be willing to give a score of weak accept, but not on its current form. I would be willing to increase my score in the next revision of the paper as long as the abstract is changed (remove the phrase "it is widely believed...") and authors add some comparison of their \textbf{techniques} with those of the other papers like Allen-Zhu et al. (which are already cited) are added in the \textbf{intro and related work}. In words, what are the technical merits of the paper compared to Allen-Zhu et al, Hanin etc? Moreover, Allen-Zhu et al Fact 7.2 is so related to this work but is not mentioned at the related work!
It is crucial to do some restructuring so that readers/researchers that are not familiar with the literature, will not to get a wrong impression/message.

===============================================================================
After authors response and their changes in the document, I change my score to 6.


**Summary Of The Paper:**

The paper's main focus is on the following question: "How the length of the output of a NN is distorted with its input length".
The main assumption is that the weights of the NN are sampled from Gaussian Distributions N(0, 2/fin) where fin is the size of the input for each particular neuron. We would like to note that this is a standard initialization that appears in the literature and implementation packages.

The authors prove upper bounds on the expected length of the output for unit length inputs where the expectation is taken with respect to the initialized. The authors also provide bounds for higher moments and prove similar results for higher dimensional manifolds (instead of 1-dimensional curves). The bound is roughly the ratio of the number of neurons of last and first layer and actually the length does not explode as the number of layers grows. This fact is claimed to be surprising by the authors (though I disagree, see my comments below).



**Summary Of The Review:**

I feel that the results technically are incremental given the literature (e.g., Allen Zhu et al "A convergence theory for deep learning via overparameterization." and other papers). Nevertheless, my main issue is the way the abstract is written (effectively overselling the result), since it is not widely believed (indeed there is a highly cited paper that claims what is "widely believed") but those researchers that have read Fact 7.2 in Allen Zhu et al do not believe this claim. Distortion on Manifolds and higher moment analysis makes the paper weak accept if my comments are addressed.

---

> ### Author Response · Authors · 2021-11-21
> **Relation to Prior Work Clarified**
>
> # Response
>
> We thank the reviewer for their careful reading, suggestions, and comments. We completely agree that it is beneficial to further detail how our results draw on past literature such as Allen-Zhu et al. and to avoid overgeneralizing what individuals in the field may believe. Thus, in addition to Sections 3.4 and 6 in which we already tried to draw out both comparisons and connections to prior work:
>
> * We have removed the phrase "widely believed" from the abstract, as suggested.
> * We have added to the Introduction, just before the statement of contributions, to indicate that prior work including Allen-Zhu et al. obtained tight bounds on the size of the input-output Jacobian and hence shows that with high probability the lengths of "infinitesimal" curves are not distorted with high probability.
> * We have added to the Related Work section to emphasize that our approach relies crucially on prior results including Fact 7.2 in the work of Allen-Zhu et al., as well adding a pointer to further discussion in Sections 3.4 and 6.
>
> Please feel free to give additional feedback on whether you feel the connections to prior literature are now made more effectively.

---

> > ### Comment · Reviewer_mxm7 · 2021-11-23
> > **Change my score to 6**
> >
> > The authors addressed my comments and I changed my score to 6.

---

### Official Review · Reviewer_VKDK · 2021-11-03

**Correctness:** 4
**Technical Novelty And Significance:** 3
**Empirical Novelty And Significance:** Not applicable
**Recommendation:** 8
**Confidence:** 4

**Main Review:**

## Strengths

The paper answers a relevant question which was previously addressed by the community albeit with wrong conclusions. The formal results and the proofs are as far as I can tell correct and they agree perfectly with the numerical simulations. The paper is very well written and easy to follow. The proofs are simple and sufficiently verbose to follow without difficulty.

The paper makes progress on understanding random neural networks which may be seen as a program analogous to that of random matrices, and is similarly likely to yield important insight and serve as a good model for real nonlinear systems.

## Weaknesses

The paper does exactly what it advertises. I can list some weaknesses but they are more "things that would be nice to have" than real problems with the current manuscript.

- One obvious question is what happens for activation functions other than the ReLU? The current manuscript relies on available estimates for ReLU networks. It would be interesting to probe this numerically and perhaps formulate some coarse conjectures.
- Another small weakness is perhaps the lack of connections to other literature such as that on Lipschitz constant of ReLU nets and injectivity of ReLU nets. There may be interesting topological implications of length and volume preservation. But again, the paper is very clear and adding too much too fast could spoil the clarity.
- While the result for curves ($d = 1$) gives (implies) matching upper and lower bounds on distortion, the result for general $d$-dimensional volumes is only an upper bound. Why is that the case? How hard would it be to get a lower bound?
- (Again somehow related to injectivity): you write in Section 4.2 that the output $\mathcal{N}(M)$ will in general be $d$-dimensional, but this seems to imply a condition on the layer widths. (E.g. a bottleneck would be problematic.) Now, since you take the expectation, even averaging $d/2$-dimensional sets will yield a $d$-dimensional set (analogously to how averaging random low-rank matrices gives a full rank matrix) and the result may hold, but I feel this merits a short discussion and perhaps a rewording.

## Notes

- The results seem to be related to bi-Lipschitzness and injectivity of ReLU networks, but not equivalent (since you calculate the $d$-dimensional Hausdorff volume). I wonder whether your techniques could indeed be used to say something about injectivity and conditioning of deep ReLU nets (perhaps along the lines of https://arxiv.org/abs/2006.08464)? A random question: what are the implications of letting the curve $M$ be a space-filling curve?
- Still related (only rephrased): the results are given for a fixed curve. If the dimensions of the layers are chosen so that some of the layers are with high probability not injective, then for a given network one can find input curves that will be highly distorted. What can be said about the totality of curves for which low distortion can be expected?
- What is the significance of the bound on $m$ in terms of the layer widths? Do you expect that similar moment bounds exist for large $m$ as well? (Related: currently you state results about the mean distortion length and moment bounds, but no high-probability result.)


### Minor things and questions

- In the introduction you write "A popular way to measure the complexity of a neural network function is to compute how it distorts lengths."—one could imagine that a network does something really bad to a curve while keeping its length intact. That would mean that the network is in some sense complex and the criterion would have to be sharpened, no?
- Section 5.1, "... than it is deep, this distortion is... " -> "the length distortion is..."
- Section 5.1, "... the constant 5/8 may need to be slightly enlarged (e.g. around (20))" -> please expand the parenthesis to make it clearer what you mean
- Proof of Theorem 5.2: "... the orthogonal projection onto this tangent space..." -> what is "this" referring to? Perhaps first define the tangent space. Also: please provide precise references for the used differential geometric results such as the transformed volume in the display before (27) or the Gram identity with wedge products in (28)
- Proof of Lemma D.2: "... except that we note that for that, given..." -> "except that given $u_1, \ldots, u_k$, there exists..."
- After this proof you write "With Lemma D.1 in hand, ...". Should that be Lemma D.2?
- Last display on page 17: should this equality be in distribution?
- Second line of the first display on page 17: I suppose the meaning of the big-O here implies (or is missing) an expectation
- page 16: "where we've used that by Lemma C.2..." -> should that be Proposition C.2? (or why not make it a Lemma?).  Please check globally for similar issues.
- In the proof of Lemma C.3, should the integral in the first display be over $B$? Also, should $B \subseteq \mathbb{S}^{n'-1}$?


**Summary Of The Paper:**

This paper shows that, given a fixed curve $M : [0, 1] \to \mathbb{R}^{n_0}$ of unit length, and a randomly-initialized (using standard He initialization) ReLU network $\mathcal{N}$ with input dimension $n_0$ and layer widths $n_1, \ldots, n_L$, the length of the curve $\mathcal{N}(M)$ is close to $\sqrt{n_L / n_0}$ on average (/with high probability). The work improves on (and corrects) related existing results for random neural networks. This existing work concludes that the length distortion grows exponentially with depth, but 1) depending on the weight normalization it is straightforward to see that this may happen; one should thus choose the standard 2/fan-in variance for initialization, 2) it contains a bug which makes it seem that even for the standard He initialization the length distortion explodes with increasing depth. The present manuscript shows (theoretically and experimentally) that this is not the case. The results make use of estimates from Giryes et al. (2016) and Hanin & Rolnick (2018) for random ReLU networks.

The main technical idea (which I find very nice) is to relate global quantities such as expected length or square-of-length to local properties of the Jacobian by using the AM-GM inequality.


**Summary Of The Review:**

This is a very well-written paper about an important topic which clears up some confusion caused by bugs in prior work. The arguments are clear and simple and likely to be useful to other researchers. Improvements and additions are certainly imaginable but inessential. I therefore recommend acceptance.

---

> ### Author Response · Authors · 2021-11-21
> **Detailed Reply to Reviewer Comments/Suggestions**
>
> # Response
> We sincerely thank the reviewer for their thorough reading, helpful comments, and variety of suggestions. We address them under the same headings as in the review.
>
> ## Weaknesses
>
> * We agree what happens for non-ReLU networks is quite an interesting question and plan to study it in future work. Significantly different techniques are required since the statistics of Jacobians in general networks cannot typically be exactly computed at finite width.
> * We agree it is natural to connect our work to literature on the Lipschitz constant of ReLU nets. We'd added a remark about to the related work section and have cited in addition related/motivating work on WGANs.
> * We agree it is natural to connect our work to literature on injectivity of ReLU networks and have added the suggested reference to the related works section.
> * It is possible to get matching lower bounds on the expectation of volume distortion for $d-$dimensional input manifolds. Doing so requires computing the moments of logarithms of $\chi^2$ random variables with a random number of degrees of freedom (a more involved version of the computation on page 17). Such things can be written in terms of the di-gamma function (the logarithmic derivative of the Gamma function). We chose not to include these computations since they are somewhat cumbersome and it wasn't clear to us that they yielded new insights.
> * We agree that averaging low-dimensional manifolds can increase dimension. However, we think the reviewer missed a point. Namely, we are computing the average Hausdorff measure of $\mathcal N(\mathcal M)$  rather than the of the Hausdorff measure of the average of $\mathcal N(\mathcal M)$. So if $\mathcal N(\mathcal M)$ has Hausdorff dimension strictly less than $d$ with probability $1$, then its $d$-dimensional Hausdorff measure vanishes with probability $1$ and hence its average is $0$. Perhaps we've misunderstood the reviewer's point and if this is the case we would appreciate a clarification.
>
> ## Notes
>
> * We agree that our work is related to conditioning and injectivity in ReLU networks but don't know how to make a precise statement. However we've added a reference to the article on globally injective ReLU networks suggested by the reviewer. Unfortunately, we can't think of anything useful that our results can say if $M$ is space filling curve.
> * It is certainly interesting to ask about what happens to the set of curves for which the length distortion is low. Our shows that for any fixed curve the probability of high distortion is low. But as the reviewer points out, the union of many low probability events can have high probability. We are not sure what can be said in this direction but thank the reviewer for the thought-provoking question.
> *At least for $1d$ curves, using Markov's inequality we can obtain high probability (though in general not exponentially high probability) bounds of the form
>     $$
>     \mathbb P\left(\mathrm{len}(\mathcal N(\mathcal M)) > t * \mathrm{len}(\mathcal M)\right) \leq t^{-m}\left(\frac{n_L}{n_0}\right)^{m/2}\exp\left[c m^2\sum_{\ell=1}^L n_\ell^{-1}\right],
>     $$
>     for any $m\geq 1$, which are contentful as soon as $t$ exceeds a large constant times $(n_L/n_0)^{1/2}\exp\left(m\sum_{\ell=1}^L n_\ell^{-1}\right)$. This is not too bad if $\sum_{\ell=1}^L n_\ell^{-1}$ is small. Moreover, we can obtain (though didn't write them down it the article) similar high probability bounds for higher dimensional input manifolds.
>
> ## Minor Things and Questions
> * We agree with the reviewer that a network can be complex without distorting the length of a given curve (or even a collection of curves). In particular, large length distortion is sufficient but not necessary to high complexity. We plan to study other notions of complexity, such as curvature distortion in future work.
> * We have fixed the typo in "... than it is deep, this distortion is ..."
> * We have added an explanation about what the constant $5/8$ needs to be potentially enlarged for general weight distributions in Sec 5.1. We had forgotten to add a kurtosis factor (which appears in our revision) to the statement given in equation (20) of the results of Hanin-Nica.
> * We have specified in the proof of Theorem 5.2 that $\Pi_{T_xM}$ is the orthogonal projection onto the tangent space $T_xM$ to $M$ at $x$.
> * We have added references for both the area formula used to derive (27) and for the Gram identity used to derive (28).
> * We have fixed the typo in Lemma D.2.
> * We have fixed the reference after the proof of Lemma D.2 to point to Lemma D.2 instead of D.1.
> * We have indicated that the last display on page 17 should be an equality in distribution.
> * We have added the expectation to the big O term on the first display in page 17.
> * We have updated the erroneous references to Lemma C.2 to refer to Proposition C.2.
> * We have corrected the display in the proof of Lemma C.3 to be an integral over $B'$.

---

> > ### Comment · Reviewer_VKDK · 2021-11-23
> > **Thanks!**
> >
> > Thank you for the various clarifications!

---

### Decision · Program_Chairs · 2022-01-20

**Decision:**

Accept (Poster)

**Comment:**

The paper studies the length distortion in a random (deep) ReLU network — namely, it bounds the expectation and higher moments of the length of the curve in feature space produced by applying a random ReLU work to a smooth curve. Because the product of layer norms grows exponentially in the depth, it might be natural to conjecture that the length grows exponentially in depth. Indeed, this has been claimed in previous theoretical work. The submission argues through rigorous mathematical analysis and corroborating experiments that this claim is incorrect. In fact, the length exhibits a slow (1/depth) contraction as the network depth increases. The paper also works out higher order moments and extensions to higher dimensional volumes. These results are obtained using nice (and natural) independence arguments and calculations.

Initial reviews were mostly positive, with the reservation that the initial submission may have slightly over claimed (the reviewer correctly notes that it is impossible to prove interesting results about the NTK of deep networks with the incorrect exponential growth hypothesis, and that related, and correct, arguments are embedded in the proofs of a number of NTK adjacent papers). After responses and revisions from the authors, the reviewers uniformly recommend acceptance. This is a solid paper, with an important conceptual point — length/volume contraction is critical to reasoning correctly about feature evolution in deep networks. In addition, it corrects existing errors in the literature, and provides relatively transparent justifications of its main claims.

The AC concurs with the reviewers’ evaluation of the paper, and recommends acceptance.